# Actinium chelation and crystallization in a macromolecular scaffold

Jennifer N. Wacker ⓘ[1], Joshua J. Woods ⓘ[1], Peter B. Rupert ⓘ[2], Appie Peterson ⓘ[1], Marc Allaire ⓘ[3], Wayne W. Lukens ⓘ[1], Alyssa N. Gaiser ⓘ[1,6,7], Stefan G. Minasian ⓘ[1], Roland K. Strong ⓘ[2] ✉ & Rebecca J. Abergel ⓘ[1,4,5] ✉

Targeted alpha therapy (TAT) pairs the specificity of antigen targeting with the lethality of alpha particles to eradicate cancerous cells. Actinium-225 [$^{225}$Ac; $t_{1/2}$ = 9.920(3) days] is an alpha-emitting radioisotope driving the next generation of TAT radiopharmaceuticals. Despite promising clinical results, a fundamental understanding of Ac coordination chemistry lags behind the rest of the Periodic Table due to its limited availability, lack of stable isotopes, and inadequate systems poised to probe the chemical behavior of this radionuclide. In this work, we demonstrate a platform that combines an 8-coordinate synthetic ligand and a mammalian protein to characterize the solution and solid-state behavior of the longest-lived Ac isotope, $^{227}$Ac [$t_{1/2}$ = 21.772(3) years]. We expect these results to direct renewed efforts for $^{225}$Ac-TAT development, aid in understanding Ac coordination behavior relative to other +3 lanthanides and actinides, and more broadly inform this element's position on the Periodic Table.

Targeted alpha therapy (TAT), a radiopharmaceutical strategy that delivers high-energy alpha ($\alpha$) particles to cancerous cells in vivo through biological targeting vectors, is a nascent approach to treat certain types of cancers[1,2]. Compared to generalized cancer treatments, TAT has shown high efficacy with minimal toxicity by directly targeting diseased cells while causing limited damage to the surrounding tissue[3,4]. The $\alpha$-emitting radioisotope actinium-225 [$^{225}$Ac; $t_{1/2}$ = 9.920(3) days] is particularly attractive for this type of radiopharmaceutical therapy given its relatively short half-life and a decay chain with multiple $\alpha$-emissions to promote an in vivo generator of $\alpha$ particles[5–7]. The potential of $^{225}$Ac-TAT has been established in clinical trials[8], such as in the treatment of advanced-stage prostate cancer[9]. As $^{225}$Ac is not naturally occurring, the burgeoning demand for this radioisotope has been met through strategic developments in isotope production[10,11]. Despite these advancements, the clinical progression of $^{225}$Ac-TAT has been slow, with $^{225}$Ac-TAT studies suffering from poor retention at the targeted site in vivo, leading to nonspecific radiotoxic effects from $^{225}$Ac

and its progeny[12,13]. Underlying these issues is the paucity of knowledge regarding the chemical behavior of this element.

Despite its discovery at the turn of the century, little research has been dedicated to understanding actinium (Ac) chemistry over the past several decades, with only a handful of studies reported with simple inorganic or organic ligands[14]. Interest in fundamental Ac coordination has experienced a revival due to the recent realization of the potential of $^{225}$Ac for TAT[15–19], yet developments are scarce as the chemistry is technically challenging, requires specialized facilities, and is currently limited to scales of micrograms or less due to the current production capabilities for $^{225}$Ac and other Ac isotopes[11,20]. Nonetheless, important advancements have been made in the theoretical and experimental gas- and solution-state chelation of Ac with synthetic and naturally occurring systems, in addition to pioneering X-ray absorption spectroscopic measurements to probe its coordination behavior in relevant solutions[21–26]. On the other hand, with no single crystal structure reported to date, our understanding of Ac

[1]Chemical Sciences Division, Lawrence Berkeley National Laboratory, Berkeley, CA 94720, USA. [2]Division of Basic Sciences, Fred Hutchinson Cancer Center, Seattle, WA 98109, USA. [3]Berkeley Center for Structural Biology, Lawrence Berkeley National Laboratory, Berkeley, CA 94720, USA. [4]Department of Nuclear Engineering, University of California, Berkeley, Berkeley, CA 94720, USA. [5]Department of Chemistry, University of California, Berkeley, Berkeley, CA 94720, USA. [6]Present address: Facility for Rare Isotope Beams, Michigan State University, East Lansing, MI 48824, USA. [7]Present address: Department of Chemistry, Michigan State University, East Lansing, MI 48824, USA. ✉e-mail: rstrong@fredhutch.org; abergel@berkeley.edu

coordination behavior in the solid-state is still limited. Foundational powder X-ray diffraction measurements, mostly from 1950[14,27], have established Ac chemical bonding concepts, but recently, the accuracy of these measurements have been scrutinized due to the possibility of impurities in the diffraction samples[28]. As such, there is a clear need to assemble a platform capable of addressing these multidimensional challenges to uncover even the most rudimentary information about this element, with particular emphasis on limiting interference from contaminants that would otherwise skew Ac bonding metrics.

Our approach to probe Ac chemical behavior was multifaceted. First, we used a multidentate, synthetic siderophore, termed 3,4,3-LI(1,2-HOPO) and referred to as HOPO hereafter, that has demonstrated efficient binding for heavy metal decorporation in vivo[29]. As HOPO is particularly amenable for chelation of lanthanide (Ln) and actinide (An) ions in aqueous solutions, we hypothesized that HOPO should bind Ac[III] analogously to other trivalent M[III] ions (M[III] = Sc, Y, La–Lu, Am, Cm, Cf, Es)[30–35], while also providing a spectroscopic handle to probe Ac[III] solution thermodynamics that would otherwise be inaccessible given the Ac[III] electronic ground state $(5f^{0}6d^{0})$[30,31]. Although HOPO can serve to examine Ac coordination in solution, an additional tactic is needed to access solid-state structural information. Macromolecular scaffolds offer an opportunity to crystallize nanogram to microgram quantities of a metal or metal-complex through high-molecular weight carriers to reveal insight for metal coordination

via protein crystallographic measurements[36]. The mammalian protein, siderocalin (Scn), binds siderophores and their ferric complexes as part of the innate immune system, and has also been shown to selectively bind a diverse range of Ln- and An-siderophore complexes[37,38]. By adding Scn to a solution containing Ac[III] and HOPO, recognition of the metal complex in the binding site of Scn will occur, and this macromolecular Ac[III]–HOPO–Scn ternary construct can be crystallized from solution. Taken together, HOPO and Scn provide a unique lens to view the fundamental coordination chemistry of Ac on a microgram scale, outlined in Fig. 1. Herein, we isolate and characterize an Ac[III]–HOPO complex contained within a macromolecular scaffold.

## Results

### Purification and quantification of an actinium stock solution

The half-life of [225]Ac, the limited availability of this isotope, and the need to possess significantly higher mass quantities to perform non-trace chemical studies were circumvented by employing the longest-lived isotope of actinium, [227]Ac [$t_{1/2}$ = 21.772(3) years], in these studies as the closest surrogate to the radiopharmaceutically-relevant isotope. The synthesis of an [227]Ac complex required a chemically and radiochemically purified stock solution attained through the reprocessing of legacy [227]Ac samples according to established methods, which combine anion exchange chromatography and extraction chromatography to remove chemical impurities and [227]Ac decay products

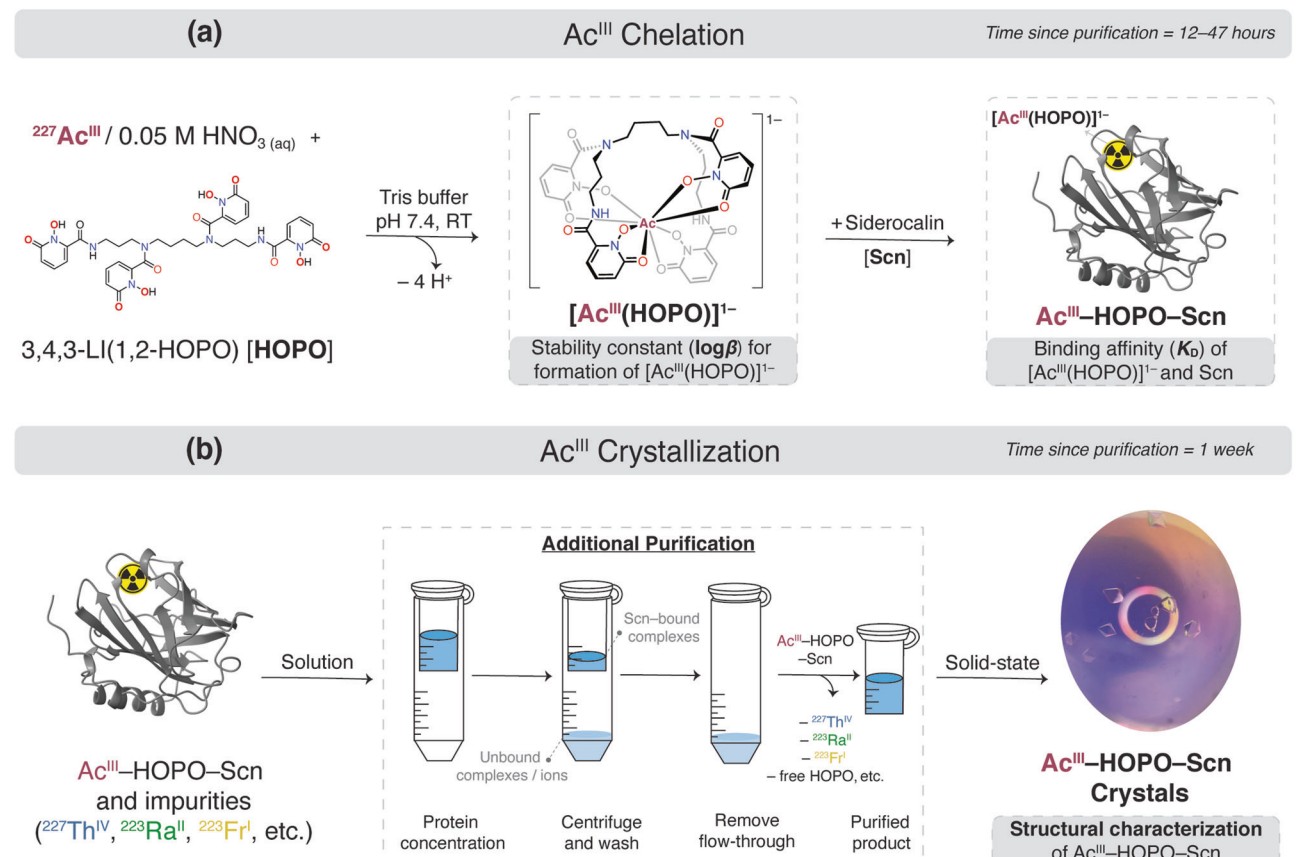

**Fig. 1 | Generalized scheme of the work presented herein that aims to understand the coordination of an actinium hydroxypyridinone complex contained within a macromolecular scaffold. a** Synthetic route to an actinium small molecule complex encapsulated in a macromolecule through chelation of [227]Ac[III] with the 3,4,3-LI(1,2-HOPO) (HOPO) chelator and the siderocalin (Scn) protein. Under buffered conditions, HOPO binds Ac[III] to yield a monoanionic coordination complex, nominally termed [Ac[III](HOPO)][1−]. Through electrostatic interactions with ligand aryl groups and amino acid residues, [Ac[III](HOPO)][1−] is then contained in the binding pocket of Scn to yield a ternary Ac[III]–HOPO–Scn protein complex. The

binding between HOPO, Ac[III], and Scn was probed using spectrofluorimetric competition titrations and fluorescence quenching binding assays. **b** The ternary Ac[III]–HOPO–Scn protein complex was crystallized to structurally characterize actinium coordination within Scn. After Ac[III]–HOPO–Scn was formed in solution, further purification was provided through protein concentration filters wherein superfluous ions, such as the radioactive daughters that had grown in since purification of the Ac[III] stock solution, were filtered out. After one week, diffraction-quality crystals of the ternary Ac[III]–HOPO–Scn complex were observed and structurally characterized with protein crystallography.

(Supplementary Fig. 1)[23–25,39]. Purifications were performed immediately before the experimental studies in order to mitigate safety concerns associated with the highly radioactive, gamma-emitting daughters of $^{227}$Ac, namely $^{227}$Th and $^{223}$Ra, and ensure the overwhelming majority of the metal stock solution contained $^{227}$Ac instead of its daughter isotopes (Supplementary Fig. 2). Further nuances arose in the quantification of the final metal stock after purification, as the low energy beta ($\beta$) and gamma ($\gamma$) emissions of $^{227}$Ac make determining its concentration directly after purification particularly challenging. As such, a combination of liquid scintillation counting to directly probe $^{227}$Ac $\beta$-emission and gamma spectroscopy to indirectly probe $^{227}$Ac with $\gamma$-emissions from daughter ingrowth were used (Supplementary Fig. 3). A thorough discussion of the purification and quantification of $^{227}$Ac is provided in the Supplementary Methods section.

## Determination of the [Ac$^{III}$(HOPO)]$^{1-}$ stability constant

Upon achieving a chemically and isotopically pure stock solution of $^{227}$Ac, the formation constant (log $\beta_{ML}$) of Ac$^{III}$ with HOPO (Fig. 1A) was determined to establish the viability of this chelator for TAT applications and understand the complexation of actinium with HOPO in the context of other +3 metal ions. With an eye towards isotope recovery and limited mass quantities, spectrofluorimetric competition titrations were chosen over other methods, such as solvent extractions or potentiometric titrations, to assess complexation thermodynamics. The high sensitivity of fluorescence spectroscopy enables thermodynamic stability constants to be determined indirectly through metal competition experiments for non-luminescent metals, like Ac$^{III}$, using europium (Eu$^{III}$) as the luminescent reference[31]. As the experiments were performed at specific pH and ionic strength values, the thermodynamic stability constants are conditional and therefore termed herein as log $\beta'_{ML}$ values instead of log $\beta_{ML}$ values. To determine the stability constant of Ac$^{III}$ with HOPO via spectrofluorimetric competition titrations with microgram quantities of Ac$^{III}$, the concentration of Eu$^{III}$ was scaled down considerably. Rather, typical competition titrations that contained 3 $\mu$M of the Eu$^{III}$ complex (in this case, [Eu$^{III}$(HOPO)]$^{1-}$) were reduced down to 10 nM, which only required approximately 1.70 $\mu$g of competing Ac$^{III}$ instead of 1.02 mg for triplicate measurements. Numerous surrogate experiments were performed separately with La$^{III}$ to support the feasibility and reproducibility of spectrofluorimetric measurements at these reduced scales (Supplementary Discussion section, Supplementary Figs. 4–8, Supplementary Table 1). Consistent log $\beta'_{ML}$ values for [La$^{III}$(HOPO)]$^{1-}$ were observed, regardless of the metal concentrations at 3 $\mu$M [Eu$^{III}$(HOPO)]$^{1-}$, 50 nM [Eu$^{III}$(HOPO)]$^{1-}$, or 10 nm [Eu$^{III}$(HOPO)]$^{1-}$ underscoring the viability of this approach for Ac$^{III}$ thermodynamic studies (Fig. 2A–C, E).

Spectrofluorimetric competition titrations between Ac$^{III}$ and [Eu$^{III}$(HOPO)]$^{1-}$ were monitored by fluorescence spectroscopy. As seen in Fig. 2D, the intensity of [Eu$^{III}$(HOPO)]$^{1-}$ decreased with the increasing concentrations of Ac$^{III}$, which corresponds to the formation of the [Ac$^{III}$(HOPO)]$^{1-}$ complex and the release of Eu$^{III}$ ions from HOPO chelators. These changes were modeled using a nonlinear least-squares refinement (Supplementary Fig. 9), and included refinement parameters such as the HOPO protonation constants measured under these solution conditions (Supplementary Figs. 10, 11, Supplementary Table 2), the hydrolysis constants of the metal ions (Supplementary Table 3), and the [Eu$^{III}$(HOPO)]$^{1-}$ formation constant[31]. In aqueous solution at pH 7.36, the conditional stability constant of the [Ac$^{III}$(HOPO)]$^{1-}$ complex was determined to be 17.0(1). This value can be compared to [La$^{III}$(HOPO)]$^{1-}$ under the same conditions with a log $\beta'_{ML}$ value of 16.50(3), or at higher concentrations with a log $\beta'_{ML}$ value of 17.08(4) (Fig. 2E). In addition, these [La$^{III}$(HOPO)]$^{1-}$ values are in agreement with a published [La$^{III}$(HOPO)]$^{1-}$ conditional stability constant collected under different solution conditions through metal competition titrations, with a reported value of 16.4(3)[31]. Generally, the conditional stability constants of Ac$^{III}$ and its non-radioactive surrogate La$^{III}$ with HOPO are lower than the rest of the lanthanides (Fig. 2F). When compared to other +3 actinides, the conditional stability constant of Ac$^{III}$ with HOPO is also lower, with [Am$^{III}$(HOPO)]$^{1-}$ and [Cm$^{III}$(HOPO)]$^{1-}$ values at 20.4(2) and 21.8(4), respectively[32,33]. More broadly, the log $\beta'_{ML}$ value determined herein for [Ac$^{III}$(HOPO)]$^{1-}$ is consistent with established bonding trends of the $f$-elements; stability constants with HOPO increase as a function of decreasing ionic radii and increasing Lewis acidity across the $4f$- and $5f$-series (Fig. 2F)[28,40].

## Chelation of [Ac$^{III}$(HOPO)]$^{1-}$ by a protein, siderocalin

Armed with solution thermodynamic data to support the complexation and formation of the [Ac$^{III}$(HOPO)]$^{1-}$ complex, the next step was to establish the recognition of this complex in solution with the protein, siderocalin (Scn). To assess the affinity of Scn for [Ac$^{III}$(HOPO)]$^{1-}$ to form the ternary complex, referred to herein as Ac$^{III}$–HOPO–Scn, the dissociation constant ($K_D$) was determined through a fluorescence quenching binding assay (Fig. 1A). The $K_D$ value can be quantified by monitoring the intrinsic fluorescence of tryptophan residues that reside in the binding cavity, or calyx, of Scn, which are thereby quenched upon metal complex recognition[37,41]. Surrogate experiments were performed with La$^{III}$ (Supplementary Discussion section, Supplementary Figs. 12–14).

A buffered solution that contained the [Ac$^{III}$(HOPO)]$^{1-}$ complex was titrated into a solution of Scn and the tryptophan emission was monitored by fluorescence spectroscopy. As seen in Fig. 3, the normalized intensity of Scn decreases with the increasing concentrations of [Ac$^{III}$(HOPO)]$^{1-}$, which is based on the recognition of the actinium complex by the protein. Fluorescence data were analyzed by nonlinear regression analysis of fluorescence response versus [Ac$^{III}$(HOPO)]$^{1-}$ concentration using a one-site binding model (Supplementary Table 4)[42]. In aqueous solution at pH 7.4, the $K_D$ of the [Ac$^{III}$(HOPO)]$^{1-}$ complex with Scn was determined to be 6(1) nM. This value can be compared to the free ligand, [HOPO]$^{4-}$, and [La$^{III}$(HOPO)]$^{1-}$, with $K_D$ values of 43(17) nM and 20(5) nM, respectively (Supplementary Tables 5, 6). Notably, the nanomolar affinity of Scn for [Ac$^{III}$(HOPO)]$^{1-}$ is significantly higher compared to HOPO and the La$^{III}$ complex (Fig. 3). Moreover, Scn shows a greater affinity for the Ac$^{III}$ complex compared to other Ln- and An-HOPO complexes, such as Eu$^{III}$, Am$^{III}$, and Cm$^{III}$ with $K_D$ values of 14(1) nM, 29(1) nM, and 22(5) nM, respectively[38]. The obtained $K_D$ value is closer to those of yttrium (Y$^{III}$) and scandium (Sc$^{III}$) HOPO complexes, [Y$^{III}$(HOPO)]$^{1-}$ and [Sc$^{III}$(HOPO)]$^{1-}$, with $K_D$ values of 2(1) nM and 6(1) nM, respectively[43]. It should be noted that the recognition of [Ac$^{III}$(HOPO)]$^{1-}$ by Scn does not quite match the avid affinity of the protein for one of its known endogenous targets, the ferric complex of enterobactin, [Fe$^{III}$(Ent)]$^{3-}$ [0.4(1) nM][37]. Nonetheless, the thermodynamically favorable formation of [Ac$^{III}$(HOPO)]$^{1-}$ coupled with the complex's binding affinity with Scn collectively generate an advantageous platform to probe actinium coordination in the solid-state.

## Crystallization of actinium in a macromolecular scaffold

The ternary Ac$^{III}$–HOPO–Scn complex was crystallized using methods previously demonstrated to routinely and effectively crystallize other +3 metal–HOPO–siderocalin macromolecules quickly (Fig. 1B)[38]. After one week, diffraction-quality crystals were isolated and analyzed at the Advanced Light Source (Beamline 5.0.2). Crystals of the ternary La$^{III}$–HOPO–Scn complex were also characterized for direct comparison between the +3 La/Ac metal centers. As demonstrated previously, Scn is capable of specific, tight binding to a wide variety of metals complexes with natural and synthetic siderophores[44]. In general, the binding of ligands/complexes in the protein calyx is driven by electrostatic and cation–$\pi$ interactions, particularly by interactions between side chains and ligand aryl groups. Although perfected to

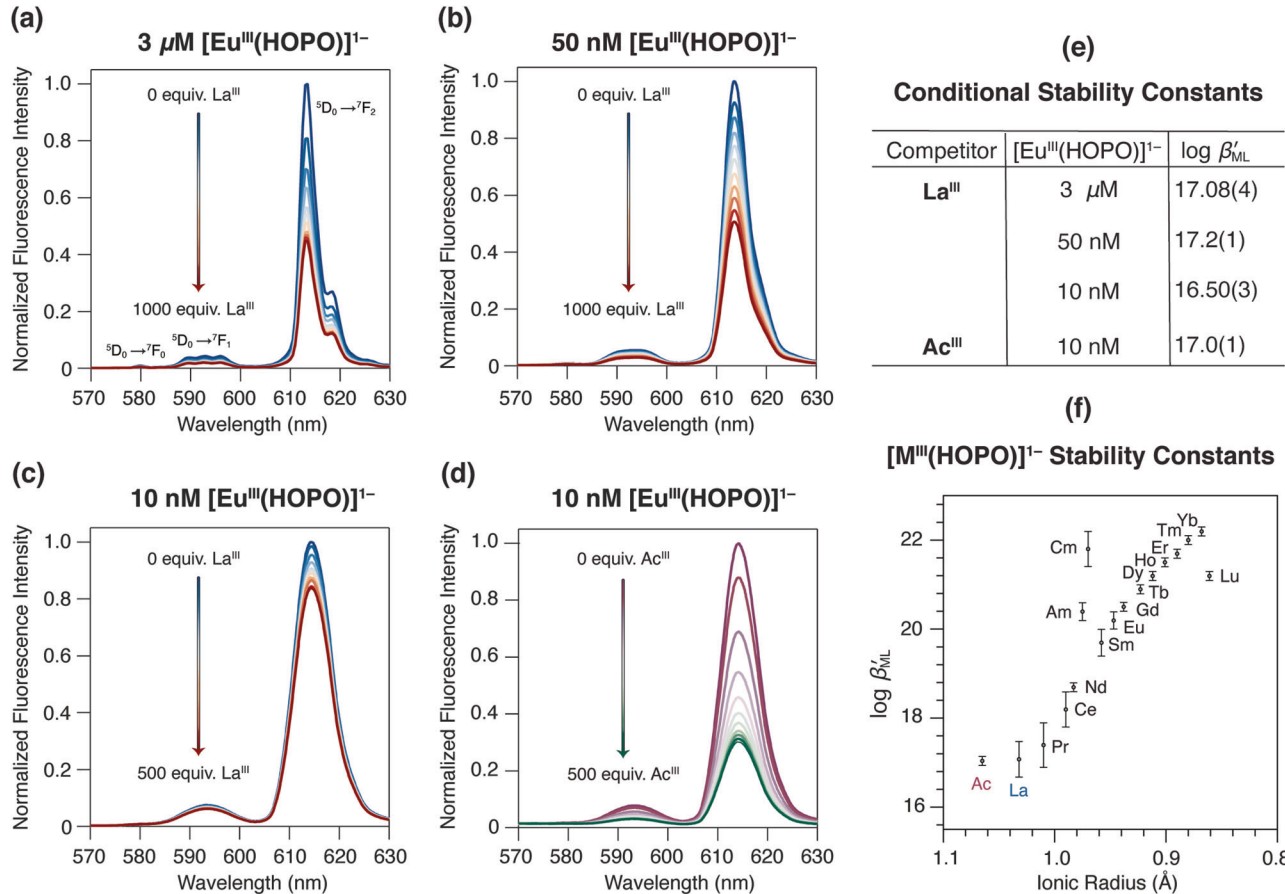

**Fig. 2 | Spectrofluorometric competition titrations used to assess the stability constants of trivalent lanthanides and actinides with the HOPO chelator. a–c, e** Surrogate experiments with La[III] were successively scaled down to repeat the experiment with limited mass quantities of Ac[III]. Competition against the [Eu[III](HOPO)]$^{1-}$ complex at concentrations of (**a**) 3 μM [Eu[III](HOPO)]$^{1-}$, (**b**) 50 nM [Eu[III](HOPO)]$^{1-}$, and (**c**) 10 nM [Eu[III](HOPO)]$^{1-}$ were performed with La[III] to demonstrate the merit of the methodology, resulting in similar conditional stability constants (log $\beta'_{ML}$) tabulated in **e**. The error reported is the standard deviation corresponding to the last digit in triplicate measurements. **d** Spectrofluorimetric competition titrations with $^{227}$Ac[III] and [Eu[III](HOPO)]$^{1-}$ resulting in a conditional stability constant (log $\beta'_{ML}$) tabulated in **e**. ([Eu[III](HOPO)]$^{1-}$ = 10 nM, $I$ = 0.5 M KCl, pH = 7.36, T = 25 °C, $\lambda_{ex}$ = 325 nm. **f** Stability constants of trivalent lanthanide and actinide ions with HOPO to yield the respective [M[III](HOPO)]$^{1-}$ complex plotted as a function of ionic radii. The log $\beta'_{ML}$ values for Ac and La are reported herein (**c**, **d**), whereas the remaining values are from refs. [31–33]. The error reported is the standard deviation corresponding to the last digit in triplicate measurements, except for americium, which is the standard deviation corresponding to the last digit in duplicate measurements. The ionic radii are from ref. [28]. (Ac) and ref. [40]. for CN = 6. Note, the stability constants increase across the 4$f$ and 5$f$ series.

accommodate hexadentate ligands like enterobactin in three pockets within the calyx, siderocalin manages to bind octadentate structures like HOPO by distorting the coordination environment of the calyx only mildly. The fourth HOPO ring of the octadentate chelator cannot fit within the calyx and rather extends outwards (Fig. 4A), yet the protein shows remarkable structural rigidity and conservation as highlighted by the structural similarities between trivalent lanthanide and actinide structures (Supplementary Fig. 15)[38,44]. The Ac[III]–HOPO–Scn structure is no different in these generalities, emulating binding in the protein scaffold akin to other $f$-elements. The most noteworthy structural rearrangement in the Ac[III] complex involves the movement of two aryl groups of the HOPO chelator, the one occupying Pocket #3 and the other extending out of the calyx, above another HOPO group, in Pocket #1 (Fig. 4A, red arrows). These movements displace the two aryl groups outwards, likely to accommodate the larger Ac[III] ion, and are not observed in the corresponding La[III] structure. Otherwise, the side-chains of residues lining the Scn calyx remain largely unperturbed, with the largest movement (0.7 Å) occuring at the Nζ position of a lysine residue (K125) that defines Pocket #1 (Fig. 4A). Further evidence of Ac[III]–HOPO–Scn crystallization were determined through qualitative radiolytic degradation of the crystals over time and quantitative radioactivity

measurements of the crystals (Supplementary Discussion section, Supplementary Figs. 16–18).

## Discussion
### Evaluation of actinium solution and solid-state metrics
Notwithstanding microgram quantities of $^{227}$Ac[III], the combination of the HOPO chelator and siderocalin enabled the collection of meaningful solution and solid-state information. The conditional stability constant of the [Ac[III](HOPO)]$^{1-}$ complex, with a log $\beta'_{ML}$ value of 17.0(1), expands the very limited library of actinium thermodynamic data[14]. Put in the context of other chelators, the [Ac[III](HOPO)]$^{1-}$ formation constant is greater than that of ethylenediaminetetraacetic acid (EDTA), with a log $\beta_{ML}$ value of 14.22 ($I$ = 0.1 M, pH = 2.8)[45], and is less than the formation constants of Ac[III] with 1,4,7,10-tetraazacyclododecane-1,4,7,10-tetraacetic acid (H$_4$DOTA) and H$_4$BATA with log $\beta_{ML}$ values of 20.32(3) and 25.74(1), respectively[19]. The lanthanide-utilizing protein lanmodulin (LanM) has been shown to recognize actinium, with a polymetallic formation constant of 36.2(5) for $^{228}$Ac$_3$LanM at pH 7, but should be noted as the product of three individual stability constants, one for each metal binding site[22]. In the same vein, by combining the affinity of siderocalin for [Ac[III](HOPO)]$^{1-}$ (measured through $K_D$), the cumulative stability constant ($\beta'_{1,2} = K_1K_2$; Supplementary Discussion

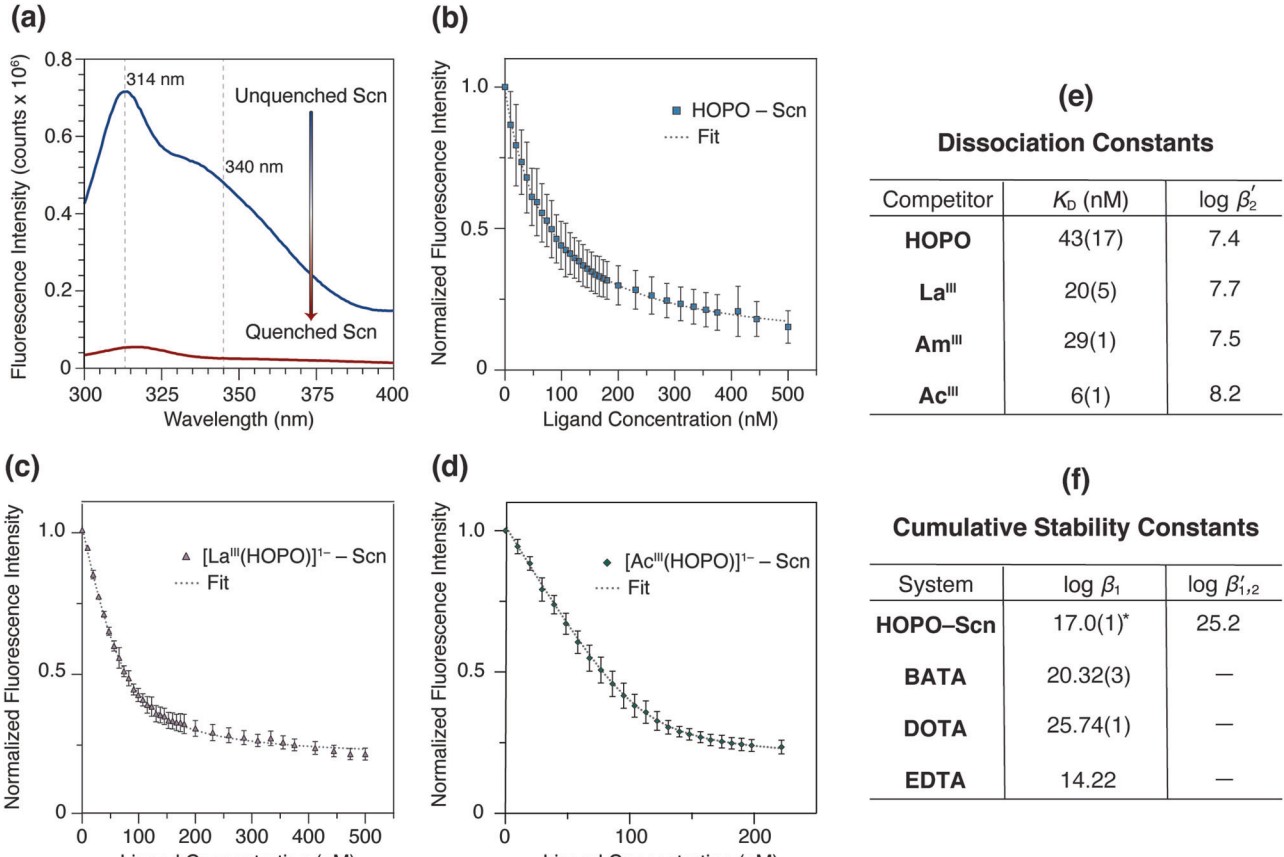

**Fig. 3 | Fluorescence quenching binding assays as a function of titrant concentration and subsequent determination of dissociation constants and cumulative stability constants. a** Luminescent behavior of tryptophan residues within the siderocalin (Scn) protein. Fluorescence ($\lambda_{ex} = 281$ nm; blue trace) is quenched upon titrant recognition in the Scn binding pocket ($\lambda_{ex} = 281$ nm; red trace), which can be used to quantify the interaction between the protein and the titrant to yield a dissociation constant ($K_D$). Fluorescence quenching analyses of Scn with HOPO (**b**), [La$^{III}$(HOPO)]$^{1-}$ (**c**), and [Ac$^{III}$(HOPO)]$^{1-}$ (**d**) at pH 7.4. Blue squares, purple triangles, and green diamonds denote the normalized fluorescence

intensities at 340 nm ($\lambda_{ex} = 281$ nm) and the error bars represent the standard deviation of three separate trials. The dotted gray fit lines were calculated with a one-binding site model[42]. **e** Tabulated $K_D$ values from fluorescence quenching analyses in **b**–**d**. The Am$^{III}$ value is from ref. 38. The log $\beta_2'$ value, which represents the binding constant of the ligand or metal complex with Scn, was calculated by taking the inverse log of the $K_D$. The error reported is the standard deviation corresponding to the last digit in triplicate measurements. **f** The cumulative stability constant (log $\beta_{1,2}'$) of HOPO–Scn with Ac$^{III}$ compared to other chelators, including DOTA[19], BATA[19], and EDTA[45]. The asterisk denotes a conditional stability constant.

**(e)**

**Dissociation Constants**

| Competitor | $K_D$ (nM) | log $\beta_2'$ |
|---|---|---|
| **HOPO** | 43(17) | 7.4 |
| **La$^{III}$** | 20(5) | 7.7 |
| **Am$^{III}$** | 29(1) | 7.5 |
| **Ac$^{III}$** | 6(1) | 8.2 |

**(f)**

**Cumulative Stability Constants**

| System | log $\beta_1$ | log $\beta_{1,2}'$ |
|---|---|---|
| **HOPO–Scn** | 17.0(1)* | 25.2 |
| **BATA** | 20.32(3) | — |
| **DOTA** | 25.74(1) | — |
| **EDTA** | 14.22 | — |

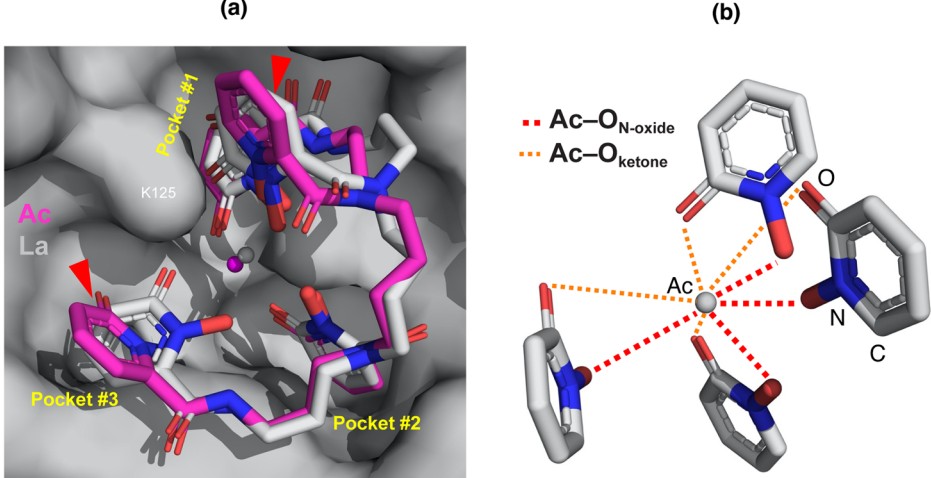

**Fig. 4 | Protein crystallographic renderings of [Ac$^{III}$(HOPO)]$^{1-}$ and [La$^{III}$(HOPO)]$^{1-}$ complexes inside the calyx of siderocalin (Scn). a** Overlay of [Ac$^{III}$(HOPO)]$^{1-}$ and [La$^{III}$(HOPO)]$^{1-}$ complexes inside the calyx of Scn. The protein is shown in a molecular surface representation (gray), the chelating groups are shown in a licorice-stick representation, colored by atom type (carbon = gray, nitrogen = blue, oxygen = red), and the metals are shown as colored spheres (Ac = magenta,

La = gray). The protein binding pockets are noted in yellow text and one of the amino acid residues (K125) that defines Pocket #1 is labeled. Red arrows highlight the positions of the HOPO groups wherein the rings are geometrically different between the La and Ac structures. **b** Representation of the interactions about the Ac$^{III}$ metal center with HOPO rings, identifying the oxygen interactions with either N-oxide or ketone oxygen atoms.

section, Supplementary Eqs. (5)–(7) for Ac$^{III}$–HOPO–Scn is 25.2, making it one of the most effective actinium chelators to date (Fig. 3F). Insight into this favorable interaction was supported through protein crystallographic data that illustrated a different geometry of the actinium complex in the protein binding cavity compared to other trivalent ions.

Herein, actinium structural characterization in the solid-state was enabled through its crystallization within a macromolecular scaffold. The structure crystallized in the $P4_12_12$ space group with a resolution of 2.08 Å (Supplementary Table 7; Supplementary Fig. 19). Three unique side chains (A, B, and C) are observed; each side chain possesses one crystallographically unique Ac$^{III}$–HOPO complex (Fig. 4A). In the complex, HOPO surrounds the Ac$^{III}$ ion in an approximate square pyramid geometry with an average Ac$^{III}$–O$_{HOPO}$ distance of 3.2(7) Å (Supplementary Table 8). In all cases, the averaged Ac–O distances involving N-oxide oxygen atoms [Ac–O$_{N\text{-}oxide}$ = 2.9(5) Å] were considerably shorter than the averaged Ac–O distances involving ketone oxygen atoms [Ac–O$_{ketone}$ = 3.5(8) Å] (Fig. 4B). These distances can be considered in the larger context of previous solution-state measurements: Ac–O$_{H2O}$ bond lengths were found to be 2.63(1) Å for the actinium-aquo ion by X-ray absorption spectroscopy (XAS) at the Ac$^{III}$ $L_3$-edge[24]. Perhaps more relevant are actinium XAS measurements with the DOTP$^{8-}$ macrocycle, wherein Ac–O$_{DOTP}$ and Ac–N$_{DOTP}$ bond lengths were found to be 2.49(1) Å and 2.87(3) Å, respectively[25]. When compared to the structure of La$^{III}$–HOPO–Scn that also crystallized in the $P4_12_12$ space group with a resolution of 2.0 Å (Supplementary Table 7; Supplementary Fig. 19), the average La–O$_{HOPO}$ distance was 2.5(2) Å, which is nearly identical to the average La–O$_{H2O}$ bond length [2.55(5) Å] in the [La$^{III}$(H$_2$O)$_9$]CF$_3$SO$_3$H solid-state structure (Supplementary Table 9)[46]. Compared to other +3 actinide–HOPO–Scn structures, the average Am$^{III}$–O$_{HOPO}$ and Cm$^{III}$–O$_{HOPO}$, are 2.5(2) Å and 2.6(2) Å, generally consistent with typical bonding trends across the 5$f$ series.(6$f$)[38] The Cf$^{III}$–O$_{HOPO}$ distance is 3.3(9) Å, which is considerably longer than typical Cf–O bonds and possibly is owed to the low resolution of this structure (2.7 Å)[34].

The caveat of protein crystallographic solid-state analyses is that the data do not typically provide the same level of atomic resolution that is obtained in small molecule crystallography. Nonetheless, protein crystallography can be a useful tool to reveal coordination chemistry, particularly with heavy elements like actinium wherein the high $Z$ is clearly distinguishable against the low $Z$ macromolecular scaffold. When supported by additional studies, the solid-state structure of Ac$^{III}$–HOPO–Scn provided structural evidence for the differences in $K_D$ values compared to its surrogate La$^{III}$, and other +3 actinides including Am$^{III}$, Cm$^{III}$, and Cf$^{III}$, with a geometrical change about the metal center adopted for Ac$^{III}$ that is not observed in the other M$^{III}$–HOPO–Scn structures. We can speculate that the larger size of actinium (Ac$^{III}$ ionic radius = 1.065 A, CN = 6)[28] enables closer interactions between the metal complex and key amino acids in the protein calyx, which are observed at longer distances in the lanthanum system (La$^{III}$ ionic radius = 1.032 A, CN = 6)[40]. For example, the two lysine residues (K125, K134) that define Pocket #1 have an average Ac···N$\zeta_{K125}$ distance of 4.0(4) Å vs. 4.5(1) Å for La···N$\zeta_{K125}$ and an average Ac···N$\zeta_{K134}$ distance of 3.9(1) Å vs. 4.5(1) Å for La···N$\zeta_{K134}$ (Supplementary Table 10; Supplementary Fig. 20). Still, the binding cavity of Scn is very rigid, and thus the HOPO coordination to Ac$^{III}$ seems to adjust in order to allow for Ac$^{III}$–HOPO recognition in the protein. To illustrate this alteration, the Ac$^{III}$–HOPO structure in Scn can be overlayed with an optimized DFT structure previously calculated for the [Ac$^{III}$(HOPO)]$^{1-}$ complex to emphasize the protein scaffold's influence on actinium coordination, for which is not observed in the La$^{III}$–HOPO–Scn structure (Supplementary Fig. 21)[47]. Future work will leverage computational analyses to tease out the impacts of siderocalin on actinium bonding. Meanwhile, these results emphasize differences in the coordination behavior of lanthanum and actinium and collectively illustrate an example wherein

the use of a non-radiological surrogate as a proxy for the radioactive ion did not fully capture their chemical complexities. As underscored by others[12,19,48], surrogate chemistry is not always sufficient to fully understand actinide behavior, and therefore efforts to continue to explore actinium chemistry, and also develop systems to tackle the unique challenges associated with working with this element, must continue to be pursued.

As is true for most of the actinide series, our fundamental understanding of actinium chemistry is severely lacking – a consequence of its rarity, difficulty of isolation, radiotoxicity, and until recently, a lack of commercial application. Even the position of this element on the Periodic Table is still debated[49]. The potential of $^{225}$Ac-TAT has since motivated the exploration of the coordination, bonding, stability, and reactivity of this element. In combining a synthetic siderophore, HOPO, to chelate Ac$^{III}$ and encapsulate the Ac$^{III}$–HOPO complex within the binding site of the siderophore-seeking protein siderocalin, solid-state structural information of an actinium complex within a macromolecular scaffold was obtained. The crystallography illustrated geometrical changes about the Ac$^{III}$ metal center that deviated from other trivalent metals, such as actinium's lanthanide surrogate, lanthanum, and heavier actinides like americium, curium and californium. The ability to compare the structural behavior of these +3 heavy metals in the solid-state captured the limitations of using surrogates in lieu of their radioactive counterparts, as the translation of their coordination chemistry may not always be identical. Taken together with the stability constant determined for the [Ac$^{III}$(HOPO)]$^{1-}$ complex, these results represent a platform that not only reveal unique Ac coordination in the solid-state, but more broadly aid in understanding periodicity across the actinide series and this element's position on the Periodic Table, while also directing renewed efforts for chelation strategies in TAT development. Future studies will focus on applying this platform to radiopharmaceutically-relevant $^{225}$Ac in vitro and in vivo studies, as we have previously demonstrated that siderocalin can be fused with antibodies and antibody fragments for the precise delivery of cargo isotopes within targeted cancer cells[43].

## Methods

All experimental details, which include general considerations, instrumentation, preparation and quantification of actinium stock solutions, potentiometric titrations, fluorescence competition semi-batch assays, fluorescence quenching assays, macromolecular crystallography details, and additional characterizations, are provided in the Supplementary Methods section in the Supplementary Information.

## Data availability

The final crystallographic models have been deposited in the Protein Data Bank (PDB) under accession codes 8UZ9 (Ac–HOPO–Scn) and 8UYN (La–HOPO–Scn). Comparisons to previously published structures were made using the following PDB and CCDC accession codes: 4ZHG (Am–HOPO–Scn), 4ZHF (Cm–HOPO–Scn), 5KIC (Cf–HOPO–Scn), 1L6M (Fe–Ent–Scn), and BUVVIX01 ([La$^{III}$(H$_2$O)$_9$]CF$_3$SO$_3$H]). All remaining data are available in the article or can be obtained from the corresponding authors upon request.

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

## Acknowledgements

This work was supported by the U.S. Department of Energy, Office of Science of the Office of Basic Energy Sciences, Chemical Sciences, Geosciences, and Biosciences Division, Heavy Element Chemistry Program at the Lawrence Berkeley National Laboratory (LBNL) under Contract No. DE-AC02-05CH11231 (WWL, SGM, RJA). Use of the Advanced Light Source at LBNL is supported by the U.S. DOE, Office of Science, Office of Basic Energy Sciences under Contract No. DE-AC02-05CH11231. Beamline 5.0.2 is supported in part by the National Institutes of Health, National Institute of General Medical Sciences,ALS-ENABLE grant P30 GM124169-01. In addition, the authors thank the following people for discussions and assistance: Stosh Kozimor and Laura Lilley for purification guidance, Katherine Shield and Jacob Branson for actinium quantification, Korey Carter, Katherine Shield, and Gauthier Deblonde for legacy sample extraction, Alexia Cosby for scientific discussions, the Lawrence Berkeley National Laboratory Radiation Protection and Environmental Health and Safety Groups for radiation and chemical protection and guidance, and Anthony Rozales, Kevin Royal, and Daniil Prigozhin for experimental assistance at the Advanced Light Source.

## Author contributions

J.N.W., J.J.W., R.K.S., and R.J.A. conceived the study and designed the experiments. J.N.W., J.J.W., A.P., A.N.G., and W.W.L. purified and quantified the actinium stock solutions. J.N.W. and J.J.W. performed the fluorescence measurements and data analyses. P.K.S. and R.K.S isolated and purified the protein. J.N.W. and A.N.G. isolated the protein crystals. J.N.W. and M.A. collected the protein datasets. P.B.R. solved the protein structures. J.N.W. and R.J.A wrote the original draft and all other authors reviewed and edited the manuscript. S.G.M., R.K.S., and R.J.A. acquired funding.

## Competing interests

R.J.A. and R.K.S. are listed as inventors on patent applications filed by LBNL and the Fred Hutchinson Cancer Center, describing inventions related to the research results presented here. The other authors declare no competing interests.
