## [Peer Review File · Nature Communications]

Actinium Chelation and Crystallization in a Macromolecular ScaffoldREVIEWER COMMENTS

Reviewer #1 (Remarks to the Author):

The authors Strong and Abergel et al. created a masterpiece of scientific work. The manuscript undoubtedly adds an important scientific value to understand the chelation chemistry of actinium in solution and in solid-state. The topic is highly timely and very relevant to the community. Although a range of Ln- and An-HOPO chelates (Inorg Chem. 2013, 52, 8805-8811) and the crystallization technique with other MIII-HOPO-siderocalin macromolecules are already published (Proc. Natl. Acad. Sci. U.S.A. 2015, 112, 10342-10347), the authors added important information within this manuscript and expanded the knowledge of actinium chemistry. By combining different methods and techniques such as obtaining crystal structures and determining the conditional stability constant of Ac-HOPO, the research work is innovative and original. The work was planned and carried out effectively using appropriate methods and techniques despite some time-consuming experiments (the reviewer appreciate it). The synthesis and purification of ^{227}Ac , the Ln/An-complexes and the macromolecular constructs are fully characterized and well-illustrated. Especially, the experimental part is well and comprehensible described. The references are adequate and timely.

The whole manuscript - from abstract to conclusion - is clearly structured, comprehensible and the analytical and experimental data presented supports the conclusions drawn in the research work.

Minor corrections:

1. It might help the reader to explain the wording “conditional” thermodynamic stability constant in the main text by explaining that it is determined indirectly through metal competition experiments.
2. Are the determined conditional stability constants of $[\text{La}(\text{HOPO})]^{1-}$ in agreement with published stability constants measured by other techniques such as potentiometric titration? If so, please add a sentence and the corresponding reference.
3. Determination of conditional formation constants: To reach the thermodynamic equilibrium it may require hours. Did the author check if the thermodynamic equilibrium was reached?
4. Analytical data of $[\text{Eu}(\text{HOPO})]^{1-}$ are missing. Please add MS spectrum/HPLC chromatogram of $[\text{Eu}(\text{HOPO})]^{1-}$ to SI.
5. Page 3; Purification and quantification of an actinium stock solution: One sentence or even listing the possible metal impurities of ^{227}Ac in the main part would help to understand the approach quickly.
6. SI, page 4, preparation of actinium stock solution: Please write the chemical name of DGA (DGA-branched resin)

7. Page 6: The last two sentence (“Collectively, these results highlight the limitations of using surrogates.....”) in the main text are really important and of fundamental importance for the community. It would be important to to shift it to the conclusion or highlight the outcome more.

Reviewer #2 (Remarks to the Author):

The manuscript describes the solution-phase thermodynamic studies and solid-state XDR study of the ^{227}Ac -HOPO complex in the Scn protein scaffold and compares Ac complexation with La and actinides. There is a large blank area in our fundamental knowledge of Ac coordination chemistry, which largely relies on speculations derived from the trends in the Periodic Table and a few reports on simple complexes and powder X-ray diffraction measurements. This is the best work so far that directly addresses the coordination chemistry of Ac with a relevant chelator, using both solution thermodynamics and single-crystal XRD in a macromolecule scaffold.

This work is highly significant – Ac is already used in clinical studies, but our understanding of this metal cation is still very poor. This report is not only fundamentally important but also highly practical. It gives a rare insight into the coordination chemistry of Ac, although limited to one specific chelator. It will inspire other work that paves the way for better chelator design and better in vivo delivery of this important radionuclide. Also, the methodology used is generally applicable to other chelators, other rarely available metal ions, and their combinations that other researchers may use to have a better understanding of the systems. To overcome the difficulties of working with material with high radiotoxicity, ultra-low quantity, and complicated decay chain, the authors designed a series of methods to tease out the thermodynamic data that are usually only doable with macro quantity material, and they validated their methods with other 3+ metal ions. The study design is sound, and the manuscript was very well written – it could be longer in the reviewer’s opinion.

Some minor comments on the manuscript: the methods and data are largely buried in the SI, and with no page limitations, the reviewer is hoping to see more details on certain parts for good reproducibility. For example, in the part on ^{227}Ac purification, what is the loading solution, and in the part on thermodynamic studies, particularly on fluorescence competition semi-batch assays, a lot more details can be given for the conditional formation constants so others can follow. Table S3 has a footnote a that the reviewer can’t find. Another comment is it may be beneficial for the authors to give some hypotheses in their discussions. The authors are very careful with their claims, which is good, but some tentative explanations of the unusual behavior would be appreciated, for example, the very high affinity of Scn to the Ac-HOPO complex, and the unusual structure of Scn-Ac-HOPO that different from other 3+ f-block metal ions.

Overall, fantastic work

Reviewer #3 (Remarks to the Author):

Wacker et al. present a study on the chelation of actinium (Ac) by a HOPO ligand and a human protein (Scn). The authors determined the stability constant for the formation of 1:1 Ac/HOPO complex through spectrofluorometric competition titrations, and more importantly, obtained successfully a crystal of a macro Ac-HOPO-Scn complex and refined its structures. The Chemistry of Ac has been largely understudied as compared to its neighboring early actinides. The findings of this work are certainly of great importance and merit publication in Nat. Commun. The authors may further improve the quality of this work by considering the following comments.

1) Is it possible to prepare a crystal of Ac-HOPO complex without participation of the protein? I notice that crystals of Ln complexes with the same HOPO ligand and other smaller HOPO ligands have been successfully prepared previously. I understand that the very limited quantities of Ac available for experiments may restrict the crystallization of the complex, but discussion of the current structural data in combination of existing crystal structures of smaller Ln/An-HOPO complexes could provide a more comprehensive picture of Ac coordination chemistry.

2) A very interesting finding in this work is the significant deviation of some of the structural parameters of Ac-HOPO-Scn complex from that of other Ln/An analogs. As already pointed out by the authors, the orientation of the hydroxypyridinone group in “Pocket 1” and “Pocket 3” is rotated away from the aryl positions in the Ac-HOPO-Scn structure (it was La-HOPO-Scn in the manuscript, should be Ac-HOPO-Scn?). Moreover, the average Ac-O(HOPO) distance of 3.2(7) Å is largely different from other Lns/Ans. These observations might be very important for understanding the unique coordination chemistry of Ac and there is a lack of deep insight in the current manuscript into these observations, although the authors have hypothesized that the interaction of the protein with Ac-HOPO is stronger than any other M(III)-HOPO structure leads to an elongation of the average Ac-O distances. It would be helpful if the authors can elucidate this finding in more detail, if possible, likely through theoretical calculations.

3) The authors determined the stability constants of Ac/HOPO through spectrofluorometric competition titrations. I am somewhat confused when looking at the spectra and the stability constants in Figure 2. As shown in Figures 2C and 2D, the variation of Eu emission intensity upon addition of the same equiv. of Ac is much more significant in Figure 2D than that upon addition of La in Figure 2C. If these spectral variations are correct, the Ac-HOPO interaction should be stronger than La-HOPO. But the stability constants obtained from fitting of these spectra are in opposite trend for Ac and La (log_bML 18 for La and 17 for Ac). The author should double check these data to correct this inconsistency.

4) In fitting of the spectral data to get the stability constants, the contribution of Th/HOPO formation was considered. Though ²²⁷Th is the first decay product of ²²⁷Ac, the amount of Th in the solution should be very limited and can be regarded negligible as compared to ²²⁷Ac, which has a half life of 21.772 years. So, I don't think it is necessary to consider the contribution of ²²⁷Th in the fitting.

Reviewer 1

- 1a. It might help the reader to explain the wording “conditional” thermodynamic stability constant in the main text by explaining that it is determined indirectly through metal competition experiments.

RESPONSE: The main text was updated to explain the meaning of “conditional” thermodynamic stability constants. See below for the updated text in red.

ACTION: *Page 3, Line 34-39...* “The high sensitivity of fluorescence spectroscopy enables **thermodynamic stability constants to be determined indirectly through metal competition experiments for non-luminescent metals, like Ac^{III}, using europium (Eu^{III}) as the luminescent reference.³¹ As the experiments were performed at specific pH and ionic strength values, the thermodynamic stability constants are “conditional” and therefore termed herein as log β'_{ML} values instead of log β_{ML} values.**”

- 1b. Are the determined conditional stability constants of [La(HOPO)]¹⁻ in agreement with published stability constants measured by other techniques such as potentiometric titration? If so, please add a sentence and the corresponding reference.

RESPONSE: The conditional stability constants of [La(HOPO)]¹⁻ have been determined previously, but only through spectrofluorimetric competition titrations in Reference 33. No potentiometric titrations of [La(HOPO)]¹⁻ have been performed. The main text has been updated to specifically call out the reported value of [La(HOPO)]¹⁻ in Reference 33. See below for the updated text in red.

ACTION: *Page 4, Line 15-18...* “**In addition, these [La^{III}(HOPO)]¹⁻ values are in agreement with a published [La^{III}(HOPO)]¹⁻ conditional stability constant collected under different solution conditions through metal competition titrations, with a reported value of 16.4(3).³¹”**

- 1c. Determination of conditional formation constants: To reach the thermodynamic equilibrium it may require hours. Did the author check if the thermodynamic equilibrium was reached?

RESPONSE: This is an important point made by the Reviewer, and is a factor we did consider in these experiments. An equilibration time of one hour for the actinium formation constant experiments was chosen to allow for equilibrium to be reached, while limiting the time allowed for the rapid ingrowth of actinium decay products. As such, the equilibration time for actinium experiments was kept as short as possible to reach equilibrium. We tested equilibration times with lanthanum, which are noted in the Supplemental Information. See text from the SI below:

“Control experiments with La^{III} provided similar values for log β'_{ML} if the solutions were allowed to equilibrate for 1 hour or 48 hours, suggesting that a 1-hour equilibration period was sufficient to achieve equilibrium (see **Table S1**)”.

ACTION: No additional action was taken.

- 1d. Analytical data of [Eu(HOPO)]¹⁻ are missing. Please add MS spectrum/HPLC chromatogram of [Eu(HOPO)]¹⁻ to SI.

RESPONSE: We thank the Reviewer for this comment. Although the [Eu(HOPO)]¹⁻ complex has been thoroughly characterized elsewhere (for example, in Reference 31), we provided the QTOF-MS spectrum, UPLC chromatogram, and a UV-vis of [Eu(HOPO)]¹⁻ in the SI for completeness.

ACTION: Pages S15-S16, Figures S5, S6, S7 were added to provided analytical data on [Eu(HOPO)]¹⁻.

- 1e. Page 3; Purification and quantification of an actinium stock solution: One sentence or even listing the possible metal impurities of ²²⁷Ac in the main part would help to understand the approach quickly.

RESPONSE: Additional information has been added in the main text to briefly summarize the purification and quantification approach. See below for the updated text in red.

ACTION: Page 3, Lines 13-26... “The synthesis of an ²²⁷Ac complex required a chemically and radiochemically purified stock solution attained through the reprocessing of legacy ²²⁷Ac samples according to established methods, **which combine anion exchange chromatography and extraction chromatography to remove chemical impurities and ²²⁷Ac decay products (Fig. S1).**^{23, 24, 25, 39} Purifications were performed *immediately* before the experimental studies in order to mitigate safety concerns associated with the highly radioactive, gamma-emitting daughters of ²²⁷Ac, **namely ²²⁷Th and ²²³Ra**, and ensure the overwhelming majority of the metal stock solution contained ²²⁷Ac instead of its daughter isotopes (**Fig. S2**). Further nuances arose in the quantification of the final metal stock after purification, as the low energy beta (β) and gamma (γ) emissions of ²²⁷Ac make determining its concentration **directly after purification** particularly challenging. **As such, a combination of liquid scintillation counting to directly probe ²²⁷Ac β -emission and gamma spectroscopy to indirectly probe ²²⁷Ac with γ -emissions from daughter ingrowth were used (Fig. S3).** A thorough discussion of the purification and quantification of ²²⁷Ac is provided in the *Supplementary Methods*.”

- 1f. SI, page 4, preparation of actinium stock solution: Please write the chemical name of DGA (DGA-branched resin)

RESPONSE: The chemical name of DGA-branched resin has been added. See below for the updated text in red.

ACTION: Page S4, Lines 3-4... “...DGA resin, branched (DGA-B, ***N,N,N',N'*-tetrakis-2-ethylhexyldiglycolamide**) to remove the ²²³Ra^{II} daughter by extraction chromatography...”

- 1g. Page 6: The last two sentence (“Collectively, these results highlight the limitations of using surrogates.....”) in the main text are really important and of fundamental importance for the community. It would be important to shift it to the conclusion or highlight the outcome more.

RESPONSE: We thank the Reviewer for highlighting the limitations of surrogates for the community and agree that it should be emphasized more in the main text. As such, adjustments have been made to expand the discussion in the Results section, in addition to highlighting this observation in the conclusions. A few updates are shown below in the red text, but the Reviewer is encouraged to read the entirety of the *Evaluation of Actinium Solution and Solid-State Metrics* and *Conclusions* paragraphs to see the full extent of the changes.

ACTION: Page 7, Line 2-8... “Meanwhile, these results emphasize differences in the coordination behavior of lanthanum and actinium and collectively illustrate an example wherein the use of a non-radiological surrogate as a proxy for the radioactive ion did not fully capture their chemical complexities. As underscored by others,^{12, 19, 48} surrogate chemistry is not always sufficient to fully understand actinide behavior, and therefore efforts to continue to explore actinium chemistry, and also develop systems to tackle the unique challenges associated with working with this element, must continue to be pursued.”

Page 7, Line 21-23... “The ability to compare the structural behavior of these +3 heavy metals in the solid-state captured the limitations of using surrogates in lieu of their radioactive counterparts, as the translation of their coordination chemistry may not always be identical.”

Reviewer 2

- 2a. The methods and data are largely buried in the SI, and with no page limitations, the reviewer is hoping to see more details on certain parts for good reproducibility. For example, in the part on ²²⁷Ac purification, what is the loading solution, and in the part on thermodynamic studies, particularly on fluorescence competition semi-batch assays, a lot more details can be given for the conditional formation constants so others can follow.

RESPONSE: The ²²⁷Ac purification loading solution has been clarified in the “*preparation of actinium stock solution*” section of the SI. In addition, additional details have been added to the “*fluorescence competition semi-batch assays*” section of the SI. We believe the details provided are sufficient for reproducibility. In addition, readers can also refer to the supplemental references that give precedent for the methods used in the actinium purifications and competition semi-batch assays.

ACTION: Additional text was inserted throughout the Supplemental Information, mainly in the “*preparation of actinium stock solution*” (Pages S3) and the “*fluorescence competition semi-batch assays*” (Pages S6-S7) sections. Changes are outlined in red text in the SI.

- 2b. Table S3 has a footnote *a* that the reviewer can't find.

RESPONSE: We thank the Reviewer for noting this oversight. There should be no reference to footnote *a* in Table S3 and the text has been edited appropriately. See updated Table S3 caption without a footnote.

ACTION: Page S27, Table S3 caption... “Table S3. Metal hydrolysis constants used in the refinement of the conditional stability constants reported herein.”

- 2c. Another comment is it may be beneficial for the authors to give some hypotheses in their discussions. The authors are very careful with their claims, which is good, but some tentative explanations of the unusual behavior would be appreciated, for example, the very high affinity of Scn to the Ac-HOPO complex, and the unusual structure of Scn-Ac-HOPO that different from other 3+ f-block metal ions.

RESPONSE: We thank the Reviewer for highlighting the limitations of our current discussion on the unusual behavior actinium with siderocalin and agree that it should be emphasized more in the main text. Although we are limited by the resolution of protein crystallography, adjustments have been made to expand the discussion in the Results section, emphasizing the differences we observed between actinium and lanthanum. We highlight both the size differences between these ions, but also the differences observed in the protein calyx when the La-HOPO vs. Ac-HOPO complex is present

(see changes to Table S10). In addition, we compared the structures in the calyx to the calculated Ac-HOPO DFT structure (see addition of Figure S20). The Reviewer is encouraged to read the entirety of the *Evaluation of Actinium Solution and Solid-State Metrics* (starting on Page 6, line 350) and *Conclusions* paragraphs to see the full extent of the changes.

ACTION: Additional text was inserted throughout the Results and Conclusions section, mainly in the *Evaluation of Actinium Solution and Solid-State Metrics* section. Changes are outlined in red text.

Reviewer 3

- 3a. Is it possible to prepare a crystal of Ac-HOPO complex without participation of the protein? I notice that crystals of Ln complexes with the same HOPO ligand and other smaller HOPO ligands have been successfully prepared previously. I understand that the very limited quantities of Ac available for experiments may restrict the crystallization of the complex, but discussion of the current structural data in combination of existing crystal structures of smaller Ln/An-HOPO complexes could provide a more comprehensive picture of Ac coordination chemistry.

RESPONSE: We agree with the Reviewer that it would be ideal to prepare a small molecule crystal of the Ac-HOPO complex without participation of the protein host. However, the maximum amount of ^{227}Ac we have access to at LBNL is between 4-5 micrograms of metal from legacy samples. In addition, the National Isotope Development Center (www.isotopes.gov, managed by the Department of Energy) does sell ^{227}Ac , but only in millicurie quantities, equating to microgram amounts of metal as well. In our experience, several hundreds of micrograms are necessary to grow diffraction-quality small molecule crystals, and therefore this approach is not feasible for actinium. As such, the protein host was absolutely necessary as it enabled us to achieve solid-state crystallographic information of this element with such small mass quantities that would otherwise not have been possible. However, we do think this is a valuable approach to compare the small molecule HOPO structures to the HOPO structures in the protein. In fact, this work is currently ongoing in our lab to compare lanthanide-HOPO small molecule structures to lanthanide-HOPO-protein structures, for which will be published in the near future. As such, these lanthanide analogues are out of the scope of this current manuscript, but the data contained herein will be referenced in this future work.

ACTION: No additional action was taken.

- 3b. A very interesting finding in this work is the significant deviation of some of the structural parameters of Ac-HOPO-Scn complex from that of other Ln/An analogs. As already pointed out by the authors, the orientation of the hydroxypyridinone group in “Pocket 1” and “Pocket 3” is rotated away from the aryl positions in the Ac-HOPO-Scn structure (it was La-HOPO-Scn in the manuscript, should be Ac-HOPO-Scn?). Moreover, the average Ac-O(HOPO) distance of 3.2(7) Å is largely different from other Lns/Ans. These observations might be very important for understanding the unique coordination chemistry of Ac and there is a lack of deep insight in the current manuscript into these observations, although the authors have hypothesized that the interaction of the protein with Ac-HOPO is stronger than any other M(III)-HOPO structure leads to an elongation of the average Ac-O distances. It would be helpful if the authors can elucidate this finding in more detail, if possible, likely through theoretical calculations.

RESPONSE: We thank the Reviewer for highlighting the limitations of our current discussion on the coordination behavior Ac-HOPO in siderocalin in contrast to other Ln/An analogs and agree that it should be emphasized more in the main text. As such, adjustments have been made to expand the discussion in the Results section, focusing on the differences we observed between actinium and

lanthanum. While we are limited by the resolution of protein crystallography, the revisions expand the current discussion of the La-HOPO vs. Ac-HOPO complexes when in the protein. In addition, we compared the La-HOPO/Ac-HOPO structures crystallized in siderocalin to the calculated Ac-HOPO DFT structure (see addition of Figure S20). Overall, the Reviewer is encouraged to read the entirety of the *Evaluation of Actinium Solution and Solid-State Metrics* (starting on Page 6, line 35) and *Conclusions* paragraphs to see the full extent of the changes to address this comment. Although we do agree that theoretical calculations would be a worthwhile pursuit to elucidate the differences observed in the Ln/An-HOPO complexes in and out of the protein, this is out of the current scope of this manuscript. However, it is the subject of ongoing work in our lab, and is now noted as such in the main text of this manuscript.

ACTION: Additional text was inserted throughout the Results and Conclusions section, mainly in the *Evaluation of Actinium Solution and Solid-State Metrics* section. Changes are outlined in red text. In particular, clarification was made for which structure contains aryl groups that are rotated away (Ac structure) and theoretical studies are now included as a focus of future work: *Page 5, Line 26-31... “The most noteworthy structural rearrangement in the Ac^{III} complex involves the movement of two aryl groups of the HOPO chelator, the one occupying Pocket #3 and the other extending out of the calyx, above another HOPO group, in Pocket #1 (Fig. 4A, red arrows). These movements displace the two aryl groups outwards, likely to accommodate the larger Ac^{III} ion, and are not observed in the corresponding La^{III} structure.”*

Page 7, Line 2-3... “Future work will leverage computational analyses to tease out the impacts of siderocalin on actinium bonding.”

- 3c. The authors determined the stability constants of Ac/HOPO through spectrofluorometric competition titrations. I am somewhat confused when looking at the spectra and the stability constants in Figure 2. As shown in Figures 2C and 2D, the variation of Eu emission intensity upon addition of the same equiv. of Ac is much more significant in Figure 2D than that upon addition of La in Figure 2C. If these spectral variations are correct, the Ac-HOPO interaction should be stronger than La-HOPO. But the stability constants obtained from fitting of these spectra are in opposite trend for Ac and La (log β_{ML} 18 for La and 17 for Ac). The author should double check these data to correct this inconsistency.

RESPONSE: We thank the Reviewer for this thoughtful comment and agree that the qualitative decrease does not match the quantitative log β'_{ML} values for [Ac^{III}HOPO]¹⁻ vs. [La^{III}HOPO]¹⁻. The 10 nM La experiments (shown in Figure 2C) were repeated in triplicate to ensure that the data represented are correct and an updated log β'_{ML} value [16.50(3)] for the 10 nM trials has been inserted in the text. The discrepancy of the initial value [18.0(7)] was likely due to limited titration points used ($n=5$). In addition, one of the trials was much higher (log $\beta'_{ML} = 18.76$) than the other two trials and likely was an outlier. By repeating this experiment with more titration points ($n=10$), the trials now look consistent between one another and the updated value (log $\beta'_{ML} = 16.5$) aligns with a previous value reported for this system in a different buffer (log $\beta'_{ML} = 16.4$; ref. 31). Nonetheless, the updated value is similar to the initial value, underscoring that despite the very small concentrations, this approach is valid to perform with actinium. In addition, the new value for [La^{III}HOPO]¹⁻ (log $\beta'_{ML} = 16.5$) is below the [Ac^{III}HOPO]¹⁻ value (log $\beta'_{ML} = 17.0$) and therefore in line with the qualitative fluorescence differences. Furthermore, the small amount of ²²⁷Th present in the ²²⁷Ac experiments can account for the additional decreases in fluorescence intensities compared to La given that the log β_{ML} for [Th^{IV}HOPO] (40.1; see reference 17 in the SI) is considerably higher than the log β'_{ML} for [Ac^{III}HOPO]¹⁻. The presence of ²²⁷Th cannot be avoided, as it grows in rapidly despite purifying the actinium stock immediately before these experiments. As such, the presence of ²²⁷Th was accounted for in the fitting of the [Ac^{III}HOPO]¹⁻ stability constant to accurately

model the contents of the solutions. For this reason, we did perform gamma spectroscopy on the solutions after each titration addition to determine the concentration of ^{227}Th throughout the experiment. Although the amount of ^{227}Th was very small (over the course of approximately 12 hours of measurements, the amount of ^{227}Th increased from approximately 1.6 picograms to 59.9 picograms), it was still included in the fit.

ACTION: The fluorescence spectra for Figures 2A-D were normalized so it is now easy for the reader to quickly look at the normalized fluorescence intensities. Furthermore, Figure 2C has been replaced with a new representative example (titrations are $n=10$ instead of $n=5$) to demonstrate the lower $\log \beta'_{\text{ML}}$ of $[\text{La}^{\text{III}}\text{HOPO}]^{1-}$ in contrast to its actinide congener, actinium. The summary table of these values has also been updated in Figure 2D.

- 3d. In fitting of the spectral data to get the stability constants, the contribution of Th/HOPO formation was considered. Though ^{227}Th is the first decay product of ^{227}Ac , the amount of Th in the solution should be very limited and can be regarded negligible as compared to ^{227}Ac , which has a half-life of 21.772 years. So, I don't think it is necessary to consider the contribution of ^{227}Th in the fitting.

RESPONSE: As the $\log \beta_{\text{ML}}$ for $[\text{Th}^{\text{IV}}\text{HOPO}]$ (40.1; see reference 17 in the SI) is considerably higher than the $\log \beta'_{\text{ML}}$ for $[\text{Ac}^{\text{III}}\text{HOPO}]^{1-}$ (17.0), it is necessary to include ^{227}Th in the fitting of the stability constants to accurately model the contents of the solutions in question. The inclusion of ^{227}Th in the fitting best represents the experimental makeup of the solutions and therefore we opted to keep it in the data reported. It also ties into the Reviewer's comment (3C) to account for the qualitative differences in the fluorescence intensity decreases between La and Ac in the competition titration experiments.

ACTION: No additional action was taken.

REVIEWERS' COMMENTS

Reviewer #1 (Remarks to the Author):

The authors have revised the manuscript extensively and have addressed all questions/comments raised by the reviewers with care and consideration. A titration experiment of $[\text{La}(\text{III})\text{HOPO}]^{1-}$ has been repeated with more data points. The results are now consistent with the conclusion drawn by the authors. In my opinion, the revised manuscript is an excellently executed and well-structured paper that is undoubtedly a real scientific asset to the community. Therefore, I highly recommend it for publication.

Reviewer #2 (Remarks to the Author):

The authors have adequately addressed the reviewers' comments with additional experiments and editing. The reviewer looks forward to seeing this important article published.

Reviewer #3 (Remarks to the Author):

I am satisfied with the authors' responses and the revised manuscript. I believe the manuscript is now in good shape for publication.